# Proline utilization A controls bacterial pathogenicity by sensing its substrate and cofactors

Peiyi Ye[1,2,3], Xia Li[1,3], Binbin Cui[1,3], Shihao Song [1,2,3], Fangfang Shen[2], Xiayu Chen[1], Gerun Wang[1], Xiaofan Zhou [2] & Yinyue Deng [1✉]

Previous reports indicate that proline utilization A (PutA) is involved in the oxidation of proline to glutamate in many bacteria. We demonstrate here that in addition to its role in proline catabolism, PutA acts as a global regulator to control the important biological functions and virulence of *Ralstonia solanacearum*. PutA regulates target gene expression levels by directly binding to promoter DNA, and its regulatory activity is enhanced by L-proline. Intriguingly, we reveal that the cofactors NAD$^+$ and FAD boost the enzymatic activity of PutA for converting L-proline to L-glutamic acid but inhibit the regulatory activity of PutA for controlling target gene expression. Our results present evidence that PutA is a proline metabolic enzyme that also functions as a global transcriptional regulator in response to its substrate and cofactors and provide insights into the complicated regulatory mechanism of PutA in bacterial physiology and pathogenicity.

[1] School of Pharmaceutical Sciences (Shenzhen), Shenzhen Campus of Sun Yat-sen University, Sun Yat-sen University, Shenzhen 518107, China. [2] Integrative Microbiology Research Center, College of Plant Protection, South China Agricultural University, Guangzhou 510642, China. [3]These authors contributed equally: Peiyi Ye, Xia Li, Binbin Cui, Shihao Song. ✉email: dengyle@mail.sysu.edu.cn

Proline is usually used as an important source of carbon and nitrogen and plays a vital role in many biological processes, such as cellular biosynthesis, cell growth, oxidative stress responses, osmotic stress responses, and redox signalling[1–4]. It can be oxidized to glutamic acid via the coordinated activity of proline dehydrogenase (PRODH), L-glutamate-γ-semialdehyde dehydrogenase (GSALDH, also named P5C dehydrogenase and ALDH4A1) and cofactors[4–9]. The enzyme that catalyses the oxidation of L-proline to L-glutamic acid is called proline utilization A (PutA) in many bacteria[7,9–11]. The domain architecture of PutA proteins indicates that they are mainly bifunctional enzymes that combine PRODH and GSALDH activity and trifunctional enzymes with an additional DNA-binding domain functioning as a transcriptional repressor of proline utilization genes[7,11]. In addition, utilization of proline by PutA was shown to contribute to the pathophysiology of some bacterial pathogens[6,12,13].

*Ralstonia solanacearum* can cause serious vascular diseases in approximately 450 kinds of plants and is responsible for one of the most devastating bacterial plant diseases worldwide[14,15]. Previous studies showed that *R. solanacearum* utilizes at least two different types of quorum sensing (QS) systems, including the methyl 3-hydroxypalmitate (3-OH PAME)[16] or methyl 3-hydroxymyristate (3-OH MAME) system[17], and the N-acylhomoserine lactone (AHL) system[18], to control important biological functions. Furthermore, the anthranilic acid signalling system also plays a vital role to control the biological functions[19]. Among these systems, both the 3-OH PAME/3-OH MAME and anthranilic acid signalling systems modulate virulence factor production, motility activity and biofilm formation[16–23].

In this study, we demonstrated that PutA not only exhibits activity as a proline utilization enzyme to convert proline to glutamate but also acts as a global responsive regulator to control the important biological functions and virulence of *R. solanacearum*. Intriguingly, the substrate proline enhances the regulatory activity of PutA to control target gene expression at the transcriptional level. Both the cofactors FAD and NAD$^+$ shift the activity of PutA from transcriptional regulation to proline catabolism. Taken together, our findings suggest the existence of an ingenious and sophisticated mechanism to switch the functions of PutA between those of a transcriptional regulator and those of a metabolic enzyme, an ability that certainly benefits the bacterial infection process.

## Results

### PutA regulates the transcription of virulence-related phenotypes of *R. solanacearum*.

To further investigate the regulatory network of *R. solanacearum* pathogenicity, we screened a Tn5 mutant library of *R. solanacearum* GMI1000 carrying a plasmid with the promoter region of the extracellular polysaccharide (EPS) -encoding gene *epsA* fused to the coding region of *lacZ* because of the critical role of EPS in *R. solanacearum* pathogenicity. We screened approximately 30,000 colonies and collected colonies that were light blue on TTC plates supplemented with X-gal. Among these identified genes (Supplementary Table 1), we focused on a gene annotated as *putA* (*RSc3301*) (Fig. 1a), which has a proline utilization A proline dehydrogenase N-terminal domain, an N-terminal DNA-binding motif, a proline dehydrogenase domain (Pro_dh domain) and a C-terminal aldehyde dehydrogenase domain (Aldedh domain) (Fig. 1b). An in-frame deletion mutant of *putA* was generated, and we then tested whether PutA regulates *epsA* at the transcriptional level. We measured *epsA* promoter-*lacZ* fusion reporter system activity in the *R. solanacearum* GMI1000 wild-type and *putA* mutant strains. Consistent with the screening result, deletion of *putA* resulted in decreased expression levels of *epsA* at various growth

stages (Fig. 1c), and this finding was subsequently confirmed by measurement of EPS production (Fig. 1d). However, the bacterial growth rates in different media did not obviously differ (Supplementary Fig. 1).

We then further studied whether the *putA* gene affects other phenotypes related to the pathogenicity of *R. solanacearum*. The motility activity and biofilm formation of the *putA* mutant were 58.32% and 32.35% lower than those of the wild-type strain, respectively, and the complemented *putA* mutant strain exhibited complete restoration of the original phenotypes (Fig. 1e, f). Consistent with these findings, the H$_2$O$_2$ sensitivity assay results showed that the survival of *putA* mutant cells was significantly decreased and was 78.27% lower than that of wild-type strain cells after treatment with 0.2 M H$_2$O$_2$ for 15 min (Fig. 1g). In addition, our results revealed that both cellulase production and the expression levels of the endoglucanase precursor (endo-1,4-beta-glucanase; cellulase) protein gene (*RSp0162*) were significantly decreased in *putA* mutant cells compared with wild-type cells (Supplementary Fig. 2).

### PutA regulates the expression of its target genes by directly binding to their promoter regions.

To determine whether transcriptional regulation of target genes is achieved by direct binding of PutA to their promoters, PutA, which has 1325 amino acids and a calculated molecular weight of 146 kDa, was purified using affinity chromatography and prepared for electrophoretic mobility shift assays (EMSA) (Fig. 2a). A PCR-amplified 336-bp DNA fragment from the *epsA* promoter was used as the probe. As shown in Fig. 2b, the *epsA* promoter DNA fragment formed a DNA-protein complex with PutA and migrated at a slower rate than unbound probe. The amount of probe bound to PutA increased with increasing amounts of PutA (Fig. 2b). And the amount of labelled probe bound to PutA was markedly decreased in the presence of a 50-fold amount of unlabelled probe (Fig. 2b). Then, we tested the binding affinity of PutA to the DNA promoter of *epsA* by microscale thermophoresis (MST). As shown in Fig. 2c, PutA bound to the *epsA* promoter DNA fragment with an estimated dissociation constant ($K_D$) of 10.73 ± 0.878 nM. Previous analysis suggested that the hydrogen bonds between Lys9 and DNA promoter contribute to the regulatory activity of PutA in *E. coil*[24]. To test the role of the lysine residue, we generated a single point mutant (K9A), in which the residue K9 was substituted by alanine (A). Whereas expression of the wild-type *putA* in the deletion *putA* mutant fully restored the phenotypes, in trans expression of the PutA derivative containing substitution at the K9 residue only partially rescued the motility activity, biofilm formation and EPS production (Supplementary Fig. 3).

To further study the binding site of the PutA protein in the *epsA* promoter, a DNase I footprinting assay was used to identify the PutA binding sequence in the promoter region of *epsA*. The PutA binding site in the probe was protected from DNase I digestion, and the binding site sequence was identified as CACTCCGAAGTAGGGAAACGAAATG (Supplementary Fig. 4a and 4b). EMSA analysis showed that no DNA-protein complexes formed when the CACTCCGAAGTAGGGAAACGAAATG fragment was deleted from the promoter region of *epsA* (Supplementary Fig. 4c), suggesting that this fragment is essential for the binding of PutA to the *epsA* promoter. To further confirm the binding site, some additional similar DNA sequences in *R. solanacearum* GMI1000 were predicted (Supplementary Fig. 5a) and were then tested by EMSA (Supplementary Fig. 5b).

### L-proline enhances the regulatory activity of PutA.

PutA was previously demonstrated to act as a bifunctional enzyme that converts proline to glutamate and to be a negative regulator of the

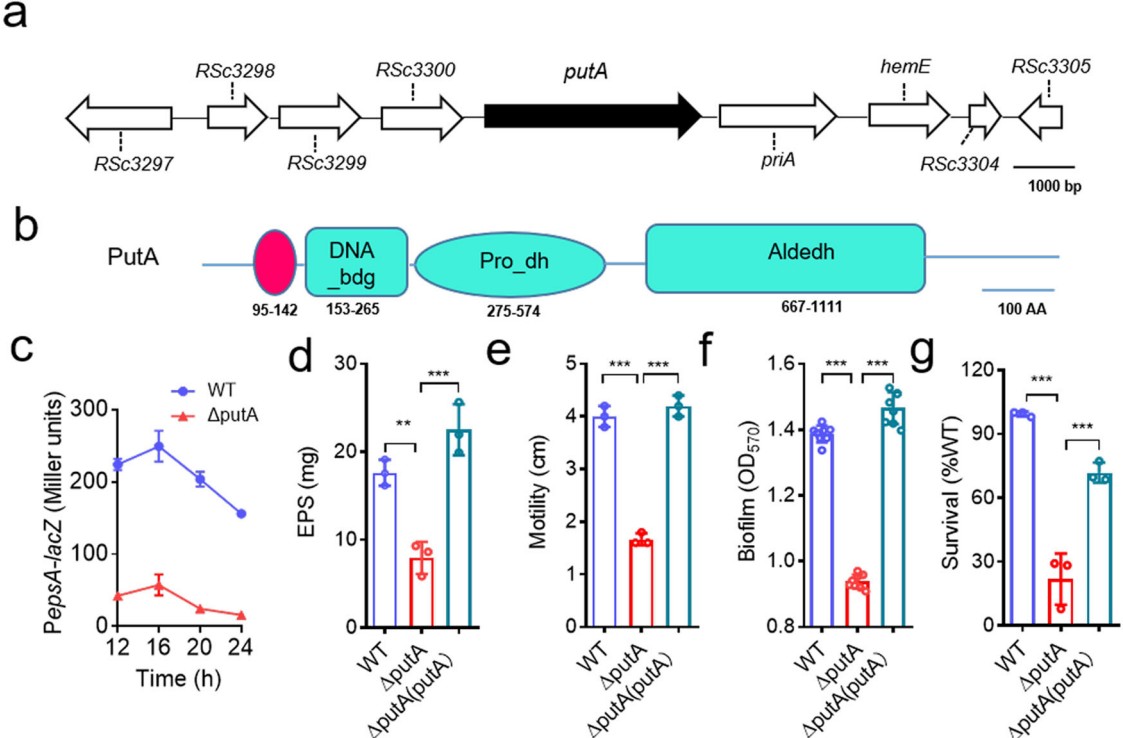

**Fig. 1 Effects of PutA on virulence-related phenotypes. a** Genomic organization of the *putA* region in *R. solanacearum* GMI1000. **b** Domain structure analysis of PutA in *R. solanacearum* GMI1000 (HMMER, https://www.ebi.ac.uk/Tools/hmmer/), the first box represents the proline utilization A proline dehydrogenase N-terminal domain (95-142). **c** Time course analysis of the effect of PutA on *epsA* gene expression, which was measured by assessing the β-galactosidase activity of the *epsA-lacZ* transcriptional fusions in the GMI1000 wild-type and *putA* mutant strains. The wild-type strain, the *putA* mutant strain, and the *putA* complemented strain were evaluated for the following virulence-related phenotypes: EPS production (**d**), motility activity (**e**), biofilm formation (**f**), and antioxidant activity (**g**). The survival rate of the *R. solanacearum* wild-type strain was arbitrarily defined as 100% and used to normalize the survival rates of the *putA* mutant and the complementation strains. Results in **c**, **d**, **e**, **f** and **g** are mean ± standard deviations of three or six independent experiments. *$p < 0.05$; **$p < 0.01$; ***$p < 0.001$ (unpaired *t*-test).

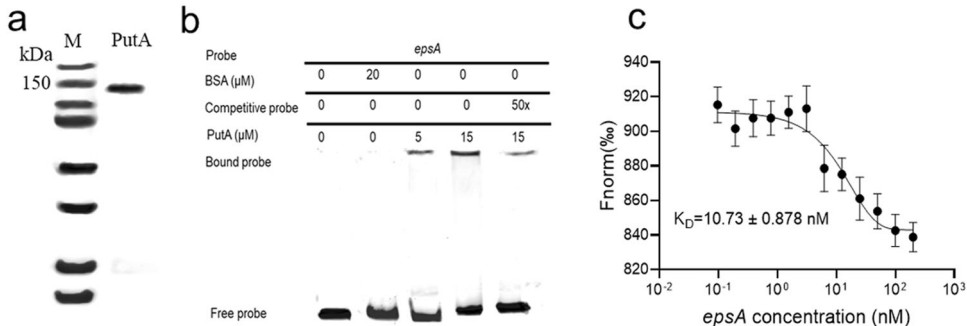

**Fig. 2 Analysis of the binding between PutA and the *epsA* promoter. a** SDS-PAGE of the purified PutA protein. **b** EMSA analysis of the binding of PutA to the *epsA* promoter in vitro. A biotin-labelled 336-bp *epsA* promoter DNA probe was used for the protein binding assay. A protein-DNA complex, represented by a band shift, was formed when different concentrations of PutA protein were incubated with the probe at room temperature for 30 min. Experiment was performed three times and representative images from one experiment are shown. **c** MST analysis of the PutA binding to *epsA* promoter DNA probe. "Fnorm (‰)" indicates the fluorescence time trace changes in MST response. Results in **c** is mean ± standard deviations of two independent experiments.

proline utilization operon, while proline switches PutA from a self-regulating transcriptional repressor to an enzymatic function in some bacterial species[9]. To investigate whether L-proline plays a role in the regulatory activity of PutA in *R. solanacearum*, we first tested the effect of L-proline on EPS biosynthesis. Exogenous addition of L-proline obviously induced the production of EPS and increased the expression levels of the *epsA* gene in a dose-dependent manner but did not obviously affect the bacterial growth rate (Figs. 3a, b, Supplementary Fig. 6). In particular, the

addition of 50 μM, 100 μM and 200 μM L-proline considerably enhanced the EPS production of *R. solanacearum* GMI1000 by 52.98%, 122.69% and 235.77%, respectively (Fig. 3b). Notably, the addition of L-proline at different concentrations did not increase *epsA* expression levels in the *putA* deletion mutant (Fig. 3a), suggesting that L-proline might affect *epsA* expression levels through PutA in *R. solanacearum*. To further explore the relationship between L-proline and the regulatory activity of PutA, we then evaluated whether L-proline affects the binding of PutA

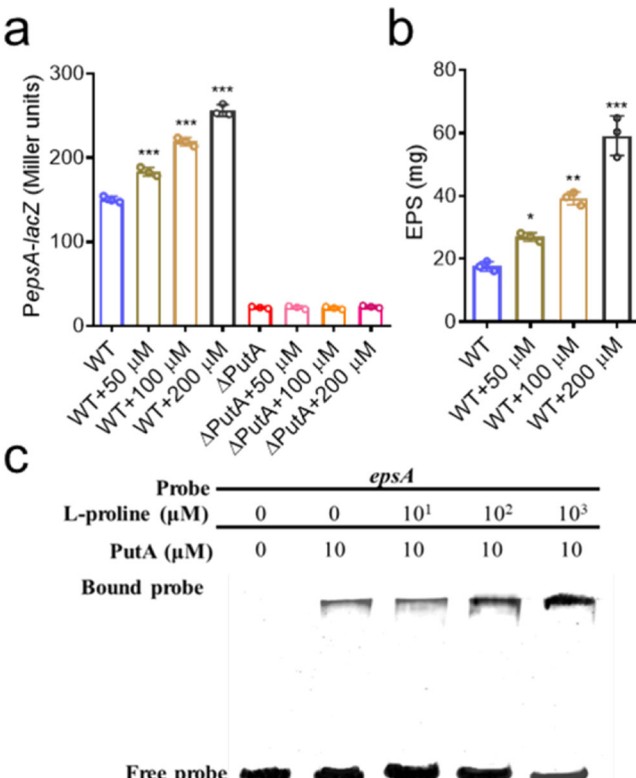

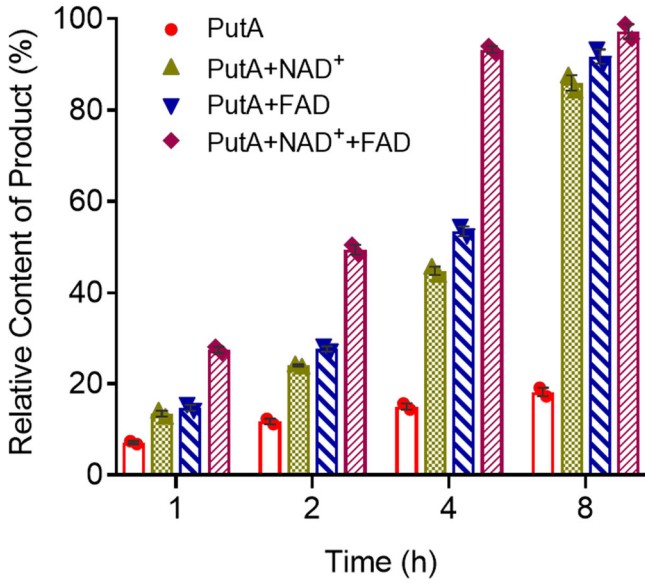

**Fig. 4 Effects of NAD$^+$ and FAD on the enzymatic activity of PutA.** The production of L-glutamate by PutA was analysed to evaluate the activity of PutA for converting L-proline with or without NAD$^+$ or FAD. The amounts of L-proline before the catalytic reactions were arbitrarily defined as 100% and used to normalize the reduction in the amount of L-proline. The data are the means ± standard deviations of two independent experiments.

**Fig. 3 Effects of L-proline on the regulatory activity of PutA. a** The effect of L-proline on *epsA* gene expression was analysed by assessing the β-galactosidase activity of the *epsA-lacZ* transcriptional fusions in the wild-type and *putA* mutant strains with the addition of different concentrations of L-proline. **b** Effect of L-proline on EPS production in the wild-type strain. **c** EMSA analysis of the effect of L-proline on the binding of PutA to the *epsA* promoter in vitro. A biotin-labelled 336-bp *epsA* promoter DNA probe was used for the protein binding assay. The protein was incubated with the probe in the presence of L-proline at different concentrations at room temperature for 30 min. Results in **a**, **b** are mean ± standard deviations of three independent experiments. *$p < 0.05$; **$p < 0.01$; ***$p < 0.001$ (unpaired *t*-test). In **c** experiment was performed three times and representative images from one experiment are shown.

to the promoter DNA of its target genes by performing EMSA. As shown in Fig. 3c, the binding of PutA to the *epsA* promoter probe was enhanced when L-proline was present in the reaction mixtures. The amount of probe bound to PutA increased with increasing L-proline concentration. However, the exogenous addition of L-glutamic acid at low concentration didn't affect the expression levels of the *epsA* gene and the production of EPS, only addition of 1 mM and 4 mM L-glutamic acid obviously enhanced the EPS production of *R. solanacearum* GMI1000 by 73.6% and 101.8%, respectively (Supplementary Fig. 7a and 7b). These results were consistently with our previous study[25]. More interestingly, L-glutamic acid did not affect the binding of PutA to the *epsA* promoter probe (Supplementary Fig. 7c).

**Both FAD and NAD$^+$ switch the activity of PutA from transcriptional regulation to proline catabolism.** PutA has a Pro_dh domain and an Aldedh domain (Fig. 1b), which exhibit PRODH and aldehyde dehydrogenase activity, respectively. We thus evaluated its enzymatic activity in vitro and found that PutA catalysed the transformation of L-proline to L-glutamic acid (Fig. 4). Interestingly, we also found that addition of the cofactors FAD and NAD$^+$ significantly enhanced the enzymatic activity of PutA for converting L-proline to L-glutamic acid (Fig. 4). We

then further investigated the effects of cofactors on the regulatory activity of PutA for controlling EPS biosynthesis. Exogenous addition of NAD$^+$ reduced the production of EPS and decreased the expression levels of the *epsA* gene in a dose-dependent manner but did not obviously affect the bacterial growth rate (Supplementary Fig. 8 and Supplementary Fig. 9). The addition of 50 μM, 100 μM and 200 μM NAD$^+$ reduced the EPS production of *R. solanacearum* GMI1000 by 8.98%, 18.22% and 30.37%, respectively (Supplementary Fig. 8b). Consistent with these results, addition of NAD$^+$ at different concentrations did not inhibit the *epsA* expression level in the *putA* deletion mutant strain (Supplementary Fig. 8a). We then performed EMSA to determine whether NAD$^+$ affects PutA binding to the promoter DNA of its target genes. As shown in Supplementary Fig. 8c, the binding of PutA to the *epsA* promoter probe was inhibited when NAD$^+$ was present in the reaction mixtures. The amount of probe bound to PutA decreased with increasing NAD$^+$ concentration (Supplementary Fig. 8c). Furthermore, we observed a similar effect of NAD$^+$ on the regulatory activity of PutA in the presence of L-proline (Supplementary Fig. 10). More interestingly, the other cofactor FAD showed the same effect as NAD$^+$ on PutA (Supplementary Fig. 11–13).

**PutA controls the signalling systems of *R. solanacearum*.** The expression of pathogenic factors in *R. solanacearum* is precisely controlled by a very complex regulatory network, in which the signalling systems play an important role. PutA controls EPS biosynthesis, cellulase production, motility and biofilm formation, which are regulated by QS systems and the anthranilic acid signalling system in *R. solanacearum*. We thus further investigated the relationship between PutA and these signalling systems. The EMSA results showed that PutA bound to DNA in the promoter regions of *trpEG*, *phcB* and *solI*, the genes encoding the synthases for anthranilic acid, the QS signals 3OH-MAME and AHL, respectively (Fig. 5a, b, c). The results of real-time quantitative reverse transcription PCR (RT-qPCR) and the promoter-*lacZ* fusion reporter assays showed that the expression levels of *trpEG*,

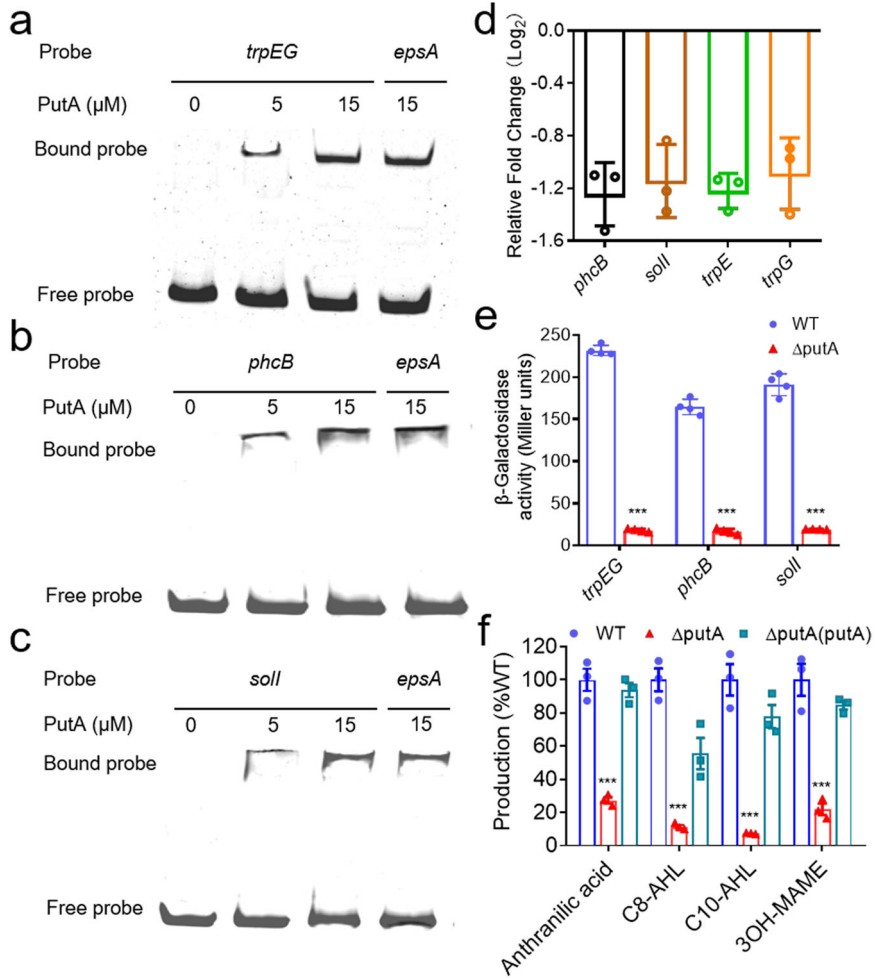

**Fig. 5 Effects of PutA on the signalling systems.** EMSA analysis of the binding of PutA to the promoters of *trpEG* (**a**), *phcB* (**b**) and *solI* (**c**) in vitro. Biotin-labelled 291-bp *trpEG*, 327-bp *phcB* and 324-bp *solI* promoter DNA probes were used for the protein binding assay. The effects of *putA* on the expression of signal synthase-encoding genes were evaluated by RT-qPCR (OD$_{600}$ = 1.0) (**d**) and by assessing the *β*-galactosidase activity of the *trpEG-lacZ*, *phcB-lacZ* and *solI-lacZ* transcriptional fusions in the wild-type and *putA* mutant strains (OD$_{600}$ = 3.0) (**e**). The signal production in the wild-type, *putA* mutant and *putA* complement strains was analysed by using UPLC-MS spectrometry. The amount of each signal in the *R. solanacearum* wild-type strain was arbitrarily defined as 100% and used to normalize the amount of that signal in the *putA* mutant and complement strains (**f**). In **a**, **b** and **c** experiment was performed three times and representative images from one experiment are shown. Results in **d**, **e** and **f** are mean ± standard deviations of three or four independent experiments. *$p < 0.05$; **$p < 0.01$; ***$p < 0.001$ (unpaired *t*-test).

*phcB* and *solI* in the *putA* mutant strain were significantly lower than those in the *R. solanacearum* GMI1000 wild-type strain (Fig. 5d, e). Consistent with these results, the production of anthranilic acid, C8-AHL, C10-AHL and 3OH-MAME was decreased significantly in the *putA* mutant strain compared with the wild-type strain (Fig. 5f). To investigate whether L-proline influences PutA regulation of the signalling systems in *R. solanacearum*, we thus tested the effect of PutA on these signalling systems in the presence of L-proline. Exogenous addition of L-proline obviously increased the expression levels of *trpEG*, *phcB* and *solI* in the wild-type strain (Supplementary Fig. 14a, b) and the production of anthranilic acid, C8-AHL, C10-AHL and 3OH-MAME was increased significantly in the wild-type strain with addition of L-proline (Supplementary Fig. 14c).

To further study the binding sites of the PutA protein in the promoters of these signal synthase-encoding genes, the similar sequences of these promoters to the PutA binding site in the promoter region of *epsA* were identified and deleted. EMSA analysis showed that no or weak DNA-protein complexes formed when these fragments were deleted from the promoter regions of *trpEG*, *phcB* and *solI* (Supplementary Fig. 15),

suggesting that these fragments are important for the binding of PutA to these promoters. MST analysis revealed that PutA bound to the *trpEG*, *phcB* and *solI* promoter DNA fragments with an estimated dissociation constant ($K_D$) of 17.14 ± 0.784 nM, 13.02 ± 0.714 nM and 12.66 ± 0.86 nM, respectively (Supplementary Fig. 16).

**PutA is self-regulating in *R. solanacearum*.** To determine whether the transcriptional expression of *putA* is self-regulated by PutA, we compared the transcript levels of *putA* in the wild-type strain and the *putA* deletion strain. Promoter activity in the mutant strain was decreased significantly compared with that in the wild-type strain, suggesting that *putA* was positively regulated by PutA at the transcriptional level (Supplementary Fig. 17a). To further study the self-regulation of PutA, we also evaluated the binding of the PutA protein to the promoter of *putA*. The EMSA results showed that in vitro, PutA can bind to its own promoter (Supplementary Fig. 17b). These results indicate that PutA not only regulates the expression of pathogenic factors of *R. solanacearum* but also controls its own expression level.

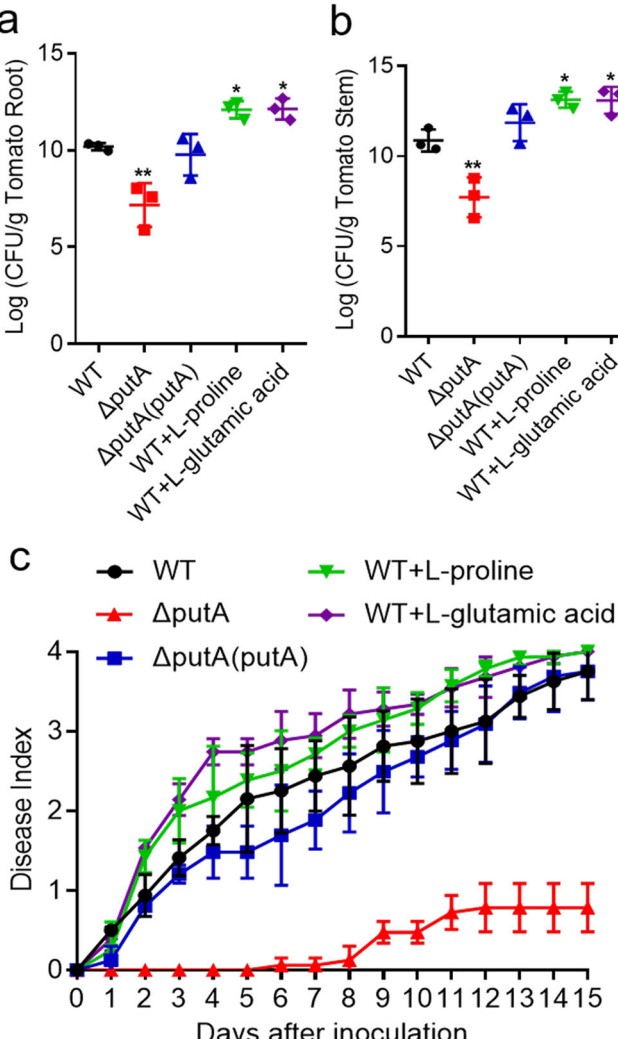

**Fig. 6 Influence of PutA on the pathogenicity of *R. solanacearum* in tomato plants.** CFUs of the *R. solanacearum* wild-type, wild-type treated with 1 mM L-proline and 4 mM L-glutamic acid, respectively, *putA* mutant and complemented strains in the roots (**a**) and stems (**b**) of tomato plants. The effects of *putA* on the virulence of *R. solanacearum* were measured by assessing the disease index of bacterial wilt in tomato (**c**). Unwounded tomato plants were subjected to soil-soak inoculation with 8 mL of bacterial culture ($OD_{600} = 1.0$) and incubated at $28 \pm 1\,^{\circ}C$ under a 12-h light/dark cycle with a relative humidity (RH) of 90%. Results in **a**, **b** and **c** are mean ± standard deviations of three biologically independent samples. *$p < 0.05$; **$p < 0.01$; ***$p < 0.001$ (unpaired *t*-test).

**PutA contributes to *R. solanacearum* pathogenicity.** Given the regulatory activity of PutA in the EPS production, cellulase production, motility, biofilm formation and antioxidant activity of *R. solanacearum*, the effect of PutA on the ability of *R. solanacearum* to infect host plants was evaluated. We first quantified the colony-forming units (CFUs) of *R. solanacearum* strains in both the roots and stems of tomato plants. The CFUs of the *R. solanacearum* wild-type, *putA* deletion and complement strains were $3.28 \times 10^{10}$, $1.49 \times 10^{7}$ and $5.8 \times 10^{9}$ per gram of root tissue, respectively, at 6 d post inoculation (Fig. 6a). A similar result was observed for tomato stems, in which the CFUs of the three strains were $1.59 \times 10^{11}$, $5.17 \times 10^{7}$ and $7.12 \times 10^{11}$ per gram of stem tissue, respectively, at 6 d post inoculation (Fig. 6b).

We further evaluated the effect of PutA, L-proline and L-glutamic acid on the pathogenicity of *R. solanacearum* in tomato plants. In good agreement with the above results, the plants treated with the *putA* mutant strain showed an obvious reduction in wilt symptoms compared with those in plants infected with the wild-type and complement strains (Fig. 6c). Wilt symptoms appeared in tomato plants beginning 1 day post infection with the wild-type GMI1000 strain, the plants treated with the wild-type strain with addition of 1 mM L-proline or 4 mM L-glutamic acid obviously increased the disease index (Fig. 6c). We also quantified the CFUs of *R. solanacearum* strains with the addition of L-proline and L-glutamic acid in both the roots and stems of tomato plants. A similar result was observed for CFUs of *R. solanacearum* strains in both the roots and stems. With the addition of L-proline and L-glutamic acid, the number of *R. solanacearum* increased significantly (Fig. 6a). These results were also consistently with the results in our previous study[25]. Deletion of *putA* significantly decreased the disease index throughout the infection process (Fig. 6c). These results suggest that PutA plays an important role in the pathogenesis of *R. solanacearum*.

**PutA controls the expression levels of a wide range of genes.** To further study the regulatory roles of *putA* in controlling bacterial biological functions, we analysed and compared the transcriptome profiles of the wild-type GMI1000 strain and the *putA* mutant strain by using RNA sequencing (RNA-seq). Differential gene expression analysis showed that the expression levels of 86 genes were increased and 92 genes were decreased ($Log_{2}$ fold-change≥1.0) in the *putA* mutant compared with their expression in the wild-type GMI1000 strain (Supplementary Fig. 18, Supplementary Table 2). These differentially expressed genes were associated with a range of biological functions, including motility, virulence, regulation, membrane components, transporters and signal transduction (Supplementary Table 2).

## Discussion

Proline plays an important role in the biological processes of many organisms[2,3]. It can be oxidized to glutamic acid by PutA in many bacteria[9,26]. In this study, we identified a PutA protein that not only exhibits enzyme activity for converting L-proline to L-glutamic acid but also controls the biological functions and virulence of *R. solanacearum* (Figs. 1, 4, 5 and 6). Previous studies reported that PutA proteins usually act as bifunctional enzymes with PRODH and GSALDH activities or as a trifunctional enzyme with an additional DNA-binding domain functioning as a transcriptional repressor of proline utilization genes[7,11]. However, to the best of our knowledge, our findings demonstrated that PutA is a global transcriptional regulator that controls EPS production, motility, biofilm formation, cellulase production and antioxidant activities (Fig. 1, Supplementary Fig. 2). We also found that PutA plays a vital role in the pathogenicity of *R. solanacearum* (Fig. 6). Taken together, the results of our study reveal a new function and role of PutA in bacterial pathogens.

Proline not only is the substrate of PutA but also switches PutA from a self-regulating transcriptional repressor to a metabolic enzyme in some bacterial species[9]. In this study, we demonstrated a different relationship between proline and PutA in *R. solanacearum*. Our results not only confirmed the enzymatic activity of PutA, which efficiently catalysed the conversion of L-proline to L-glutamic acid, in vitro (Fig. 4) but also indicated that proline obviously induced EPS production and *epsA* expression in a dose-dependent manner by enhancing the binding of PutA to the *epsA* promoter (Fig. 3). Our study provides new insight into the role of L-proline in the pathogenicity of *R. solanacearum*, in which it modulates the regulatory activity of PutA to control the production of pathogenic factors.

FAD and NAD$^+$ are cofactors of PRODH and GSALDH, respectively, which coordinately convert proline to glutamic acid in two sequential oxidative steps. We found that both FAD and NAD$^+$ also play an important role in the transformation of L-proline to L-glutamic acid in *R. solanacearum*. Addition of both FAD and NAD$^+$ significantly promoted the enzymatic activity of PutA in vitro (Fig. 4). Intriguingly, exogenous addition of either FAD or NAD$^+$ decreased the expression level of *epsA* and EPS production by reducing the binding of PutA to the *epsA* promoter region in both the absence and presence of L-proline (Supplementary Fig. 8, 10, 11, 13). Our results identified the functional switching of PutA between the different conditions of the presence and absence of FAD and NAD$^+$ and provide the evidence confirming the role of FAD and NAD$^+$ in transcriptional regulation in bacteria by modulating the regulatory activity of PutA. However, the mechanism by which FAD and NAD$^+$ affect PutA needs further investigation.

QS signals are widely used by bacteria to regulate biological functions in response to cell density. Previous studies showed that *R. solanacearum* uses two different types of QS systems: a *phc* QS system with 3-OH PAME and 3-OH MAME as signals and a typical *lux* QS system dependent on AHL signals. Our recent study identified a new signalling system, the anthranilic acid system, which controls important biological functions and the production of 3-OH MAME and AHL signals in *R. solanacearum* GMI1000[19]. In this study, we found that PutA positively controls the production of 3-OH MAME, AHL and anthranilic acid signals by directly controlling the transcriptional levels of the signal synthase-encoding genes by binding to the similar sequences in these promoters (Fig. 5, Supplementary Fig. 15, 19). We also found that the Lys9 residue is crucial to the regulatory function of PutA as in trans expression of the PutA derivative containing substitution at this site only partially rescued the phenotypes (Supplementary Fig. 3). This result is consistent with the previous study which suggested that the hydrogen bonds between Lys9 and DNA promoter affect the regulatory activity of PutA in *E. coil*[24]. Sequence alignment analysis revealed that the PutA of *R. solanacearum* GMI1000 shares a high homology with the PutA of *E. coli*, and the two PutA proteins possess the conserved site Lys9 (Supplementary Fig. 20). Taken together, our findings suggest that PutA not only directly controls target gene expression but also controls the signalling network to modulate target gene expression, suggesting the complicated hierarchy of the regulation of physiology and virulence in *R. solanacearum*. In addition, the BLAST search with the PutA homologies revealed that PutA is highly conserved in many bacteria (Supplementary Table 3), suggesting that the regulatory mechanism of PutA might be present in various bacterial pathogens.

## Methods

**Bacterial strains and culture conditions**. The bacterial strains and plasmids used in this work are listed in Supplementary Table 4. *R. solanacearum* strains were cultured in TTC medium (10 g/L tryptone, 5 g/L D-glucose, 1 g/L casein hydrolysate) or on TTC agar (TTC medium containing 15 g/L agar) at 28 °C[27]. *Escherichia coli* strains that can be used for general cloning and conjugal transfer were cultured in Luria-Bertani medium (10 g/L tryptone, 5 g/L yeast extract, 5 g/L NaCl; pH 7.0-7.5) or on LB agar (LB medium containing 15 g/L agar) at 37 °C. In the EPS quantitative experiment, SP medium (5 g/L peptone, 20 g/L glucose, 0.5 g/L KH$_2$PO$_4$ and 0.25 g/L MgSO$_4$) was used to culture *R. solanacearum*[19]. All media were supplemented with antibiotics according to the experimental requirements. The following antibiotics were used in this work: gentamicin (50 μg/mL), tetracycline (10 μg/mL), kanamycin (100 μg/mL) and ampicillin (100 μg/mL). L-Proline (HPLC ≥ 99%) and L-glutamic acid (HPLC ≥ 99%) were dissolved in ddH$_2$O to a final concentration of 100 mM, and this solution was added to the medium in the experiments.

**Construction of reporter strains and the Tn5 transposon mutant library**. The specific primers used for PCR amplification of the promoter fragments of *phcB*, *solI* and *putA* are listed in Supplementary Table 5. The promoter fragments were

inserted upstream of the promoterless *lacZ* gene in pME2-*lacZ*. The *phcB*, *solI*, and *putA* reporter plasmids were transformed into *R. solanacearum* GMI1000 by electroporation to obtain the reporter strains, which were used to determine β-galactosidase activity[28].

A mini-Tn5 transposon with a gentamicin resistance gene was transformed into *R. solanacearum* GMI1000 expressing the *epsA* promoter-*lacZ* fusion by electroporation. The transformants were selected on TTC plates supplemented with X-gal, tetracycline and gentamicin. Light blue colonies were selected for the identification of inserted sites. High-efficiency thermal asymmetric interlaced PCR was used to identify DNA flanking sequences at the insertion site of the Tn5 transposon[29].

**Construction of in-frame deletion mutants and complementation**. Gene knockout was achieved by DNA homologous recombination. *R. solanacearum* GMI1000 was used as the parental strain[30]. For complementation analysis, the coding regions of *putA* were amplified by PCR and cloned into the vector pLAFR3 under the control of the *lac* promoter and introduced into the deletion mutant strain by using electroporation. The primers used for knockout, complementation and overexpression are listed in Supplementary Table 5.

**Bacterial growth analysis**. Overnight bacterial cultures in TTC medium were washed twice in fresh TTC medium, SP medium and MP minimal medium and inoculated into the corresponding fresh medium to an optical density at 600 nm (OD$_{600}$) of 0.1[19]. A 200 μL aliquot of the cell suspension was added to each well at 28 °C in a low-intensity shaking model using the Bioscreen-C automated growth curve analysis system. The media were used as blank controls. MP minimal medium (1 L) was composed of the following constituents: FeSO$_4$·7H$_2$O, $1.25 \times 10^{-4}$ g; (NH$_4$)$_2$SO$_4$, 0.5 g; MgSO$_4$·7H$_2$O, 0.05 g; KH$_2$PO$_4$, 3.4 g. The pH was adjusted to 7.0, 20 mM glutamic acid was added.

**Biofilm formation analysis**. Overnight cultures were diluted to an OD$_{600}$ of 0.1 with TTC medium, added to the 96-well polystyrene plates and grown in static culture for 20 hours at 28 °C. The culture media were then poured out of the wells, and the wells were stained with crystal violet for 15 min and washed three times with water before the addition of 95% ethanol. Biofilm formation was quantified by measuring the absorbance at 570 nm.

**Motility activity analysis**. Motility activity was determined on semi-solid agar (0.3%)[19]. Bacteria were inoculated into the centre of plates containing 1% tryptone (Becton, Dickinson and Company, Maryland, USA) and 0.3% agar (Becton, Dickinson and Company, Maryland, USA). The plates were incubated at 28 °C for 48 h before the diameter of the colonies was measured[31].

**The H$_2$O$_2$ analysis**. Three millilitres of overnight culture (OD$_{600}$ = 1.0) was collected, washed twice with sterile PBS and resuspended in sterile PBS with 0.2 M H$_2$O$_2$. The suspension was cultured for 15 min at 28 °C and diluted in a gradient. Then, the diluted suspensions were plated on TTC plates to quantify the bacterial CFUs. CFUs of *R. solanacearum* were quantified to analyse oxidative stress sensitivity[32].

**EPS analysis**. To quantify the production of EPS[19], 100 mL of overnight culture (OD$_{600}$ = 2.5) was centrifuged at 8000 rpm for 20 min. The collected supernatants were mixed with a 4-fold volume of 95% ethanol, and the mixture was stored at 4 °C overnight. The precipitated EPS was isolated by centrifugation and dried overnight at 55 °C before determination of the dry weight.

**Protein purification and analysis**. The PutA protein expression vector was constructed, the coding region of PutA was amplified with the primers listed in Supplementary Table 5 and fused to the expression vector PDBHT2. The resulting construct was transformed into *E. coli* strain BL21 (DE3) which was cultured in LB containing the kanamycin at 37 °C. When the culture medium reached an OD$_{600}$ of 0.6, isopropyl b-D-thiogalactopyranoside (IPTG) was then added to the medium to a final concentration of 1 mM and incubated at 16 °C for 10 h. Affinity purification of the HIS-PutA fusion protein was using the His Trap affinity columns (GE Healthcare, Connecticut, USA) according to the manufacturer's instructions. Fusion protein cleavage with TEV Protease (Beyotime, Shanghai, China) was conducted at 4 °C overnight. The purified protein was eluted and verified by SDS-PAGE.

**Microscale thermophoresis assay**. Protein-binding experiments were carried out with Nano Temper 16 Monolith NT.115 instrument (NanoTemper Technologies; www.nanotemper-technologies.com)[33]. In brief, PutA protein was labelled with the L014 Monolith NT.115 Protein Labelling Kit (Nano Temper, Munich, Germany). The final concentration of labelled PutA is 50 nM, while the concentration of the promoters of *epsA*, *solI* and *phcB* is 200 nM, and the concentration of the promoter of *trpEG* is 250 nM. Labelled PutA protein and titres of unlabelled promoter DNA fragment were mixed and loaded onto standard treated silicon capillaries (k022

Monolith NT.115, Nano Temper, Munich, Germany) and fluorescence was measured. The measurements were carried out at 40% LED power and 40% MST power.

**Quantitative analysis of QS signal production**. *R. solanacearum* GMI1000 wild-type and *putA* mutant strains were grown in TTC medium overnight with agitation at 28 °C, respectively. One liters of culture supernatant was collected by centrifugation and extracted with equal volume of ethyl acetate. The crude extract (organic phase) was dried using a rotary evaporator and dissolved with methanol. All above samples were kept at 4 °C until analysis. Ultra-high-performance liquid chromatography-electrospray ionization tandem mass spectrometry (UHPLC-ESI-MS/MS) was performed in a Shimadzu LC-30A UHPLC system with a Waters C18 column (1.8 μm, 150 × 2.1 mm) and a Shimadzu 8060 QQQ-MS mass spectrometer with an ESI source interface. The mass spectrometer was operated in positive-ion mode. The mobile phase was prepared as 0.1% formic acid/water and acetonitrile.

**DNase I footprinting assay**. 450 ng of probe was incubated with 0 and 30 μg of PutA protein at 25 °C for 30 min, and 10 μL of Dnase I (0.015 unit, Promega, USA) was added for further incubation at 37 °C for 1 min. Then, 140 μL of Dnase I stop solution (200 mM unbuffered sodium acetate, 30 mM EDTA and 0.15% SDS) was used to terminate the reaction. Samples were extracted with phenol/chloroform and precipitated with ethanol. Samples were then analysed in an ABI GeneScan 500 LIZ instrument (Thermo, Waltham, USA)[34].

**Real-time quantitative reverse transcription PCR**. *R. solanacearum* cells were cultured to an OD$_{600}$ of 1.0 and were then harvested. RNA was isolated using an Eastep Super Total RNA Extraction Kit (Promega, Madison, USA). cDNA synthesis and RT-qPCR analysis were performed with ChamQ$^{TM}$ Universal SYBR qPCR Master Mix (Vazyme, Nanjing, China) according to the manufacturer's instructions in a 7300Plus Quantitative Real-Time PCR System. As a control, the expression of the 16 S rRNA gene was analysed by RT-qPCR. The relative expression levels of the target genes were calculated using the comparative CT ($2^{-\Delta\Delta CT}$) method[19].

**Pathogenicity assays**. Naturalistic soil-soak assays and tomato plant infection models were used for pathogenicity assays[25]. In brief, each plant was inoculated by pouring 8 mL of bacterial suspension (OD$_{600}$ = 1.0) into the soil. Inoculated tomato plants were maintained under 16-hour light/8-hour dark conditions at 28 ± 1 °C in a greenhouse. The inoculation experiments were repeated three times, with eight plants per group in each experiment. The disease status of tomato plants was assessed daily by scoring the disease index on a scale of 0–4[25]. All plants were monitored for disease index analysis, and the following scale was used: 0, no symptoms; 1, 1–25% wilted leaves; 2, 26–50% wilted leaves; 3, 51–75% wilted leaves; and 4, 76–100% wilted leaves. At 5 d post inoculation, 1 g of tissues from the plant roots and stems was collected, milled in a sterile mortar with 9 mL of sterile water and diluted in a gradient. Then, the diluted suspensions were plated on TTC plates to quantify the bacterial CFUs in the tomato roots and stems.

**Electrophoretic mobility shift assay**. EMSA was performed according to the instructions for the Thermo Fisher Scientific kit with minor modifications[35]. In brief, the purified PCR products of the promoters were 3-end labelled with biotin by using Thermo Fisher Scientific's Biotin 3' End DNA Labeling Kit. Protein-DNA binding interactions were detected by using a LightShift Chemiluminescent EMSA Kit. DNA-protein binding reactions were performed according to the manufacturer's instructions (Thermo Fisher, Waltham, USA). A 5% polyacrylamide gel was used to separate the DNA-protein complexes. After UV cross-linking, the biotin-labelled probes were detected in the membrane using a biotin luminescence detection kit (Thermo Fisher, Waltham, USA).

**Assessment of the enzymatic activity of PutA**. The L-proline degradation activity of PutA was determined in assays using PutA (5 μM) and NAD$^+$ (200 μM) or FAD (100 μM) in phosphate-buffered saline (PBS) with fixed concentrations of L-proline (1 mM) and coenzyme Q1 (CoQ1, 240 μM). The reaction solution was collected after 1, 2, 4 and 8 h at 28 °C, boiled for 10 min, filtered through the sample tube after centrifugation, and analysed by UPLC-MS spectrometry[36].

**Statistical and reproducibility**. Statistical analyses were performed with Graph-Pad Prism 8. The data are presented as the means ± standard deviations. Asterisks in figures indicate corresponding statistical significance as it follows: *$p < 0.05$; **$p < 0.01$; ***$p < 0.001$ (unpaired *t*-test).

**Reporting summary**. Further information on research design is available in the Nature Research Reporting Summary linked to this article.

## Data availability
Source data is provided as Supplementary Data 1. The raw figures of this article are provided as Supplementary Data 2. All other data are available from the corresponding author upon reasonable request (Lead contact: dengyle@mail.sysu.edu.cn).

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

## Acknowledgements

This work was supported by grants from the Science, Technology and Innovation Commission of Shenzhen Municipality (No. JCYJ20200109142416497), the National Natural Science Foundation of China (No. 31571969) and the National Key Research and Development Program of China (2021YFA0717003).

## Author contributions

Y.D. designed the research. P.Y., X.L., B.C., S.S. and F.S. performed the research. P.Y., X.L., B.C., S.S., F.S., X.C., G.W., X.Z. and Y.D. analysed the data. P. Y., X.L. and Y.D. wrote the paper.

## Competing interests

The authors declare no competing interests.
