## [Peer Review File · Communications Biology]

Reviewers' comments:

Reviewer #1 (Remarks to the Author):

Ye et al. describe the potential roles in PutA in *Ralstonia solanacearum*. However, it is difficult to agree with the conclusions of this paper for the following reasons. Therefore, I cannot recommend this paper for publication in *Communication Biology*.

1. As a prerequisite, the intracellular and extracellular concentrations of L-proline and L-glutamate under the experimental conditions are unknown, so the validity of the experimental design cannot be determined.
2. The authors have previously proposed that anthranilic acid may be a QS signaling molecule, but this has not been properly proven to be the case. Therefore, it is necessary to rewrite the abstract and introduction section mainly.
3. In Fig 3ab. Related to concern 1, I cannot determine the validity of the concentration of L-proline that was processed. Also, it should have been mandatory to test for L-glutamate as well.
4. In Fig 5def. It is necessary to examine how the treatment with L-proline or L-glutamate changes the respective genes and compound amounts.
5. In Fig 6. Treatment of plants with L-proline or L-glutamate is required during pathogenicity tests. If this is difficult, co-treatment with these compounds along with the deficient strains is recommended.

Reviewer #2 (Remarks to the Author):

In the manuscript "Proline utilization A controls bacterial pathogenicity by sensing its substrate and cofactors", Ye et al. show that in *Ralstonia solanacearum* PutA, in addition to being a catabolic proline enzyme and a repressor of its own synthesis, can also act as a transcriptional regulator of gene expression to control biological functions and virulence in *R. solanacearum*. The regulative functions of PutA are exerted by direct binding to target promoters and are modulated by proline, NAD⁺, and FAD, which switch PutA from a catabolic enzyme to a transcription factor.

While the general features of *R. solanacearum* PutA are of limited and specialized interest, the finding that PutA can control virulence and growth acting as a global regulator of transcription is novel and of wider interest.

A relative weakness of this manuscript, though, is the lack of experiments *in vivo* (ChIP) supporting the binding of PutA to the promoters of EPE and PutA.

The claim that PutA, besides being a catabolic enzyme, is a global transcriptional regulator is convincing but only based on *in vitro* methods, which, by nature, could hide possible artifacts, especially when a single protein is challenged with a single DNA. Furthermore, the level of purification of PutA protein is not well determined and the PutA protein still contains a histidine TAG raising the doubt that trace contaminants or the presence of the TAG itself can alter the results. As to the histidine tag, there are several reports in the literature describing interfering effects of histidine tags on DNA binding properties or competitive effects between cofactors and histidine binding (see for example Majorek et al (2014). *Protein Science*, 23, 1359-1368 or Paul et al. (2020) *Protein expression and purification*, 167, 105541.

Similarly, but more importantly, the claim that PutA is a global regulator of transcription would be much more convincing if supported by an *in vivo* whole-genome technique such as a ChIP-on-chip.

A third point I am perplexed about is relative to the DNA binding site of PutA identified in the *eps* promoter. The claim that putA is a global transcriptional regulator implies the presence of putA binding sites in the target genes. Accordingly, the identification of the binding site in the promoters of the QS synthase genes would strongly reinforce the EMSA data opening the way for a thorough screening for other target genes (if any) controlled by PutA.

Minor concerns

Line 22. Usually looks redundant. I suggest removing it

Line 63. Add a space after "(2-OH PAME)"

Line 64 and 65. Add a space after "system"

Line 84. Define the acronym EPS at its first occurrence

Line 98. What is the complement strain? You should define it somewhere (i.e. in material and methods or supplementary information)

Line 99 Replace finding (singular) with findings (plural)

Line 145. See comment of line 22

Line 332. Correct the spelling of "supplementary information" 
Line 351 Incomplete reference

Line 353 This reference needs some editing

Reviewer #3 (Remarks to the Author):

The authors tackle an important aspect of proline metabolism by seeking to identify genes critical for infection of plants by the pathogen *Ralstonia solanacearum*. New findings that potentially could advance the field are the discovery that PutA is a global transcriptional regulator including genes responsible for synthesis of quorum sensing molecules. Deletion of the putA gene provides strong support for PutA/proline having a key role in the pathogenesis of *R. solanacearum* in tomato plants. Indeed, the diminished disease index of the putA mutant strain is remarkable. Another important finding is the identification of a potential PutA binding site in the promoter region of the gene responsible for EPS biosynthesis. Despite these interesting findings, there are a number of issues with the study that weaken the conclusions.

1. Line 80, PutA is not a new transcriptional regulator. Perhaps change this section title to "PutA regulates the transcription of virulence-related phenotypes of *R. solanacearum*"

2. Line 99, change "finding" to "findings"

3. Line 109, change "assay" to "assays"

4. Line 133, remove "However"

5. The authors show evidence that expression of target genes are repressed in the putA mutant strain, e.g., the QS molecules in Figure 5. These results suggest that PutA is a transcriptional activator instead of a transcriptional repressor, which is typically how PutA regulates gene expression. On line 182, the authors also report that PutA is a positive regulator of the putA gene. What is the mechanism of PutA activating gene expression? Is it proposed to enhance binding of RNA polymerase? Based on the proximity of the PutA binding site, it seems that PutA binding to the DNA could also block RNA polymerase from recognizing the promoter region.

6. Figure 1, provide residue numbers for the three domains of PutA in the illustration.

7. Figure 2b, in the EMSAs, what was the concentration of the DNA probe? It seems strange that not all of the probe was bound at the high concentrations of PutA, namely 10-20 μ M. Also, the competitor DNA did not completely abolish PutA DNA binding. These assays should be repeated in the presence of non-specific DNA or a probe that lacks the binding sequence identified in the DNAase1 footprinting assays. This reviewer is concerned that non-specific binding is occurring between PutA and the probe DNA.

8. On Line 464 and elsewhere the authors state "imaging was performed three times" What does it mean that imaging was performed three times? Was it meant instead to say that the experiments were conducted three times?

9. Figure 3c, It is difficult to appreciate the changes in bound DNA with proline even though proline does seem to impact the reporter gene expression. Could the authors add a plot of the ratio of bound/free probe vs [proline]? Perhaps this would show the proline dependence better?

10. Figure 4, it is not unexpected to see an increase in PutA enzyme activity with FAD and NAD

since these cofactors are a required part of the reaction. What percent of the purified PutA already contains bound FAD? Why is adding FAD to the assays necessary?

11. Figure 5, similar to the other EMSAs, it is not clear why PutA does not bind all of the probe DNA in these assays. Do the DNA probes in these experiments share a similar sequence to that of the promoter region for the *epsA* promoter?

12. Proline is reported to increase PutA-DNA binding whereas FAD and NAD⁺ are reported to decrease PutA binding to DNA. All of these factors are required for PutA enzymatic activity, what regulatory mechanism are the authors proposing? It is unclear. What about examining the ratio of NAD⁺/NADH, that would test whether PutA is a transcriptional redox sensor.

13. What is the binding constant of PutA to the DNA? This is critical information and would help determine whether there is a priority of PutA binding to the different promoter regions of the target genes. Also, what change in binding affinity induced by proline or NAD⁺ would be sufficient to have meaningful regulatory effects?

14. Did the authors find any proline transporter genes in the global transcription analysis? It seems likely that PutA would also be involved in regulating proline transport.

Point-to-point response to reviewers' suggestions for COMMSBIO-21-1704-T

Reviewers' comments:

Reviewer #1 (Remarks to the Author):

Ye et al. describe the potential roles in PutA in *Ralstonia solanacearum*. However, it is difficult to agree with the conclusions of this paper for the following reasons. Therefore, I cannot recommend this paper for publication in Communication Biology.

1. As a prerequisite, the intracellular and extracellular concentrations of L-proline and L-glutamate under the experimental conditions are unknown, so the validity of the experimental design cannot be determined.

Response: In our previous study, we revealed that L-glutamic acid promotes virulence factor production and pathogenicity of *R. solanacearum* GMI1000 only at high concentrations (1- 4 mM), and the hybrid sensor histidine kinase/response regulator RS01577 is involved in the L-glutamic acid signaling in *R. solanacearum* (Shen *et al.*, *Molecular Plant Pathology*, 2020). In this study, intriguingly, we found that L-proline enhances the regulatory activity of PutA to control target gene expression and pathogenicity of *R. solanacearum* GMI1000 at 50-200 μ M (Fig. 3, Fig. 6, Fig. S13). Consistently, we also provided one more evidence to confirm the effect of L-glutamic acid on the virulence factor production and pathogenicity of *R. solanacearum* GMI1000 only at high concentrations of 1 or 4 mM, but it exerted no effect at the final concentrations of 100 or 200 μ M (Fig. 6, Fig. S6). Furthermore, we found that L-glutamic acid exhibits no effect on the regulatory activity of PutA to control target gene expression (Fig. S6). These findings show that the regulatory mechanism of L-proline is different from that of L-glutamic acid in *R. solanacearum*, supporting the novelty and significance of our findings in this study.

We also measured the intracellular and extracellular concentrations of L-proline and L-glutamic acid as suggested. As shown in the table, these results will definitely not affect the quality and conclusions of our findings in this study.

	L-proline (intracellular)	L-proline (extracellular)	L-glutamic acid (intracellular)	L-glutamic acid (extracellular)
Wild-type strain	9.17 μ M	4.15 μ M	185.48 μ M	156.15 μ M
putA mutant strains	107.9 μ M	13.14 μ M	99.08 μ M	98.5 μ M
SP Medium	N/A	1.31 μ M	N/A	51.84 μ M

2. The authors have previously proposed that anthranilic acid may be a QS signaling molecule, but this has not been properly proven to be the case. Therefore, it is necessary to rewrite the abstract and introduction section mainly.

Response: Good suggestion, we have revised these sentences as suggested (Page 2, Line 29; Page 3, Line59; Page 8, Line162; Page 8, Line165; Page 12, Line246).

3. In Fig 3ab. Related to concern 1, I cannot determine the validity of the concentration of L-proline that was processed. Also, it should have been mandatory to test for L-glutamate as well.

Response: Good suggestion, we have measured both the intracellular and extracellular concentrations of L-proline and L-glutamic acid as suggested. We have also tested the effects of L-glutamic acid and found that L-glutamic acid enhances virulence factor production and pathogenicity of *R. solanacearum* GMI1000 only at high concentration of 1 or 4 mM, but it exerts no effect on *R. solanacearum* GMI1000 at the final concentration of 100 or 200 μ M (Fig. 6, Fig. S6). Furthermore, we found that L-glutamic acid exhibits no effect on the regulatory activity of PutA to control target gene expression (Fig. S6). These findings suggest that the regulatory mechanism of L-proline is different from that of L-glutamic acid in *R. solanacearum*.

Fig.3

Fig. S6

4. In Fig 5def. It is necessary to examine how the treatment with L-proline or L-glutamate changes the respective genes and compound amounts.

Response: Good suggestion. We have added some more experiments to response to this concern. First, we have tested the effect of L-proline on the signal production and the expression level of the synthase-encoding genes (Fig. S13). Second, we have predicted and confirmed the PutA binding sites on the promoters of *trpEG*, *phcB* and *soII* (Fig. S14). Third, we have also tested the effects of L-glutamic acid and found that L-glutamic acid enhances virulence factor production and pathogenicity of *R. solanacearum* GMI1000 only at high concentration of 1 or 4 mM, but it exerts no effect on *R. solanacearum* GMI1000 at the final concentration of 100 or 200 μ M (Fig. 6, Fig. S6). Furthermore, we found that L-glutamic acid exhibits no effect on the regulatory activity of PutA to control target gene expression (Fig. S6). These findings suggest that the regulatory mechanism of L-proline is different from that of L-glutamic acid in *R. solanacearum*.

Fig. S13

Fig. S14

5. In Fig 6. Treatment of plants with L-proline or L-glutamate is required during pathogenicity tests. If this is difficult, co-treatment with these compounds along with the deficient strains is recommended.

Response: Good suggestion, we have added the experiments as suggested (Fig. 6).

Reviewer #2 (Remarks to the Author):

In the manuscript "Proline utilization A controls bacterial pathogenicity by sensing its substrate and cofactors", Ye et al. show that in *Ralstonia solanacearum* PutA, in addition to being a catabolic proline enzyme and a repressor of its own synthesis, can also act as a transcriptional regulator of gene expression to control biological functions and virulence in *R. solanacearum*. The regulative functions of PutA are exerted by direct binding to target promoters and are modulated by proline, NAD⁺, and FAD, which switch PutA from a catabolic enzyme to a transcription factor.

While the general features of *R. solanacearum* PutA are of limited and specialized interest, the finding that PutA can control virulence and growth acting as a global regulator of transcription is novel and of wider interest.

A relative weakness of this manuscript, though, is the lack of experiments in vivo (ChIP) supporting the binding of PutA to the promoters of EPE and PutA.

The claim that PutA, besides being a catabolic enzyme, is a global transcriptional

regulator is convincing but only based on in vitro methods, which, by nature, could hide possible artifacts, especially when a single protein is challenged with a single DNA. Furthermore, the level of purification of PutA protein is not well determined and the PutA protein still contains a histidine TAG raising the doubt that trace contaminants or the presence of the TAG itself can alter the results. As to the histidine tag, there are several reports in the literature describing interfering effects of histidine tags on DNA binding properties or competitive effects between cofactors and histidine binding (see for example Majorek et al (2014). Protein Science, 23, 1359-1368 or Paul et al. (2020) Protein expression and purification, 167, 105541.

Response: Good suggestion, we have cleaved the tag from the PutA protein and repeated all the relevant experiments as suggested (Fig. 2, Fig. 3, Fig. 5, Fig. S7, Fig. S9, Fig. S10, Fig. S15).

Similarly, but more importantly, the claim that PutA is a global regulator of transcription would be much more convincing if supported by an in vivo whole-genome technique such as a ChIP-on-chip.

Response: Good suggestion. We have added the experiments to analyze and compare the transcriptome profiles of the wild-type GMI1000 strain and the *putA* mutant strain by RNA sequencing (RNA-seq) (Table S2, Fig. S16), and found that the expression levels of 178 genes changed, including motility, virulence, regulation, transcriptional regulators, membrane components, transports and signal transduction. These results suggest that PutA is a global regulator to control various gene expression in *R. solanacearum*.

A third point I am perplexed about is relative to the DNA binding site of PutA identified in the *eps* promoter. The claim that *putA* is a global transcriptional regulator implies the presence of *putA* binding sites in the target genes. Accordingly, the identification of the binding site in the promoters of the QS synthase genes would strongly reinforce the EMSA data opening the way for a thorough screening for other target genes (if any) controlled by PutA.

Response: Good suggestion. To confirm the PutA binding site (CACTCCGAAGTAGGGAAACGAAATG) in the promoter region of *epsA*, we first deleted this fragment from the promoter region of *epsA* and EMSA analysis showed that there were no DNA-protein complexes formed, suggesting that this fragment is essential for the binding of PutA to the *epsA* promoter (Fig. S3C). Second, to further confirm the binding site, four additional similar DNA sequences in *R. solanacearum* GMI1000 were predicted (Fig. S4a) and were then tested by EMSA (Fig. S4). Third, the similar DNA sequences of the PutA binding site in the promoter region of *epsA* were predicted in the promoter regions of *trpEG*, *phcB* and *soll*, and were then deleted. EMSA analysis showed that there were no or very weak DNA-protein bindings between PutA and the mutated promoter probes, suggesting that these fragments are very important for the binding of PutA to these promoters (Fig. S14).

Minor concerns

Line 22. Usually looks redundant. I suggest removing it

Response: We have deleted it as suggested.

Line 63. Add a space after "(2-OH PAME)"

Response: We have added a space as suggested.

Line 64 and 65. Add a space after "system"

Response: We have added it as suggested.

Line 84. Define the acronym EPS at its first occurrence

Response: We have defined it as suggested.

Line 98. What is the complement strain? You should define it somewhere (i.e. in material and methods or supplementary information)

Response: Good suggestion, we have added more details in methods as suggested.

Line 99 Replace finding (singular) with findings (plural)

Response: We have revised it as suggested.

Line 145. See comment of line 22

Response: We have deleted "usually" as suggested.

Line 332. Correct the spelling of "supplementary information"

Response: We have revised them as suggested.

Line 351 Incomplete reference

Response: We have revised it as suggested.

Line 353 This reference needs some editing

Response: We have modified all the references as suggested.

Reviewer #3 (Remarks to the Author):

The authors tackle an important aspect of proline metabolism by seeking to identify genes critical for infection of plants by the pathogen *Ralstonia solanacearum*. New findings that potentially could advance the field are the discovery that PutA is a global transcriptional regulator including genes responsible for synthesis of quorum sensing molecules. Deletion of the putA gene provides strong support for PutA/proline having a key role in the pathogenesis of *R. solanacearum* in tomato plants. Indeed, the diminished disease index of the putA mutant strain is remarkable. Another important finding is the identification of a potential PutA binding site in the promoter region of the gene responsible for EPS biosynthesis. Despite these interesting findings, there are a number of issues with the study that weaken the conclusions.

1. Line 80, PutA is not a new transcriptional regulator. Perhaps change this section title to "PutA regulates the transcription of virulence-related phenotypes of *R. solanacearum*"

Response: We have revised it as suggested.

2. Line 99, change "finding" to "findings"

Response: We have revised it as suggested.

3. Line 109, change "assay" to "assays"

Response: We have revised it as suggested.

4. Line 133, remove "However"

Response: We have revised it as suggested.

5. The authors show evidence that expression of target genes are repressed in the putA mutant strain, e.g., the QS molecules in Figure 5. These results suggest that PutA is a transcriptional activator instead of a transcriptional repressor, which is

typically how PutA regulates gene expression. On line 182, the authors also report that PutA is a positive regulator of the putA gene. What is the mechanism of PutA activating gene expression? Is it proposed to enhance binding of RNA polymerase? Based on the proximity of the PutA binding site, it seems that PutA binding to the DNA could also block RNA polymerase from recognizing the promoter region.

Response: The binding position of transcriptional regulators on promoter DNA usually determines their functions in bacterial cells. In most cases, the binding position of bacterial transcriptional regulators on the upstream of RNA polymerase recognition core region exert transcriptional activation effect, while the binding position of bacterial transcriptional regulators on the middle or downstream of RNA polymerase recognition core region exert transcriptional inhibition effect (Feng et al., 2016, *Science*, Liu et al., 2017, *Science*). The -35 region (TTGACA) of prokaryotic promoter is the recognition site of RNA polymerase σ factor, and the -10 region (TATAAT) is the binding site of RNA polymerase to DNA. We identified the binding regions of PutA to the promoter regions of *epsA*, *trpEG*, *phcB* and *soll* are all on the upstream of RNA polymerase recognition.

6. Figure 1, provide residue numbers for the three domains of PutA in the illustration.

Response: Thanks for the good suggestion, the residue numbers for the three domains of PutA have been added in Fig. 1 as suggested.

7. Figure 2b, in the EMSAs, what was the concentration of the DNA probe? It seems strange that not all of the probe was bound at the high concentrations of PutA, namely 10-20 μ M. Also, the competitor DNA did not completely abolish PutA DNA binding. These assays should be repeated in the presence of non-specific DNA or a probe that lacks the binding sequence identified in the DNAase1 footprinting assays. This reviewer is concerned that non-specific binding is occurring between PutA and the probe DNA.

Response: Good suggestion. The concentration of the DNA probe was 3 ng/ μ L. We have repeated all the EMSA assays. We have also used BSA protein to test the

non-specific DNA binding, and the binding sequences of *epsA*, *trpEG*, *phcB* and *soll* were also removed to further confirm the bindings between PutA and the probes are specific (Fig. S3, Fig. S14).

8. On Line 464 and elsewhere the authors state "imaging was performed three times" What does it mean that imaging was performed three times? Was it meant instead to say that the experiments were conducted three times?

Response: It means that the experiments were conducted three times. We have revised them as suggested.

9. Figure 3c, It is difficult to appreciate the changes in bound DNA with proline even though proline does seem to impact the reporter gene expression. Could the authors add a plot of the ratio of bound/free probe vs [proline]? Perhaps this would show the proline dependence better?

Response: Good suggestion, the plots of the bound/free probe vs proline are shown in the following Figure. The Figure a represents the amount of free probe in EMSA, while the Figure b represents the amount of bound probe in EMSA.

10. Figure 4, it is not unexpected to see an increase in PutA enzyme activity with FAD and NAD since these cofactors are a required part of the reaction. What percent of the

purified PutA already contains bound FAD? Why is adding FAD to the assays necessary?

Response: The purified PutA didn't contain FAD (data not shown). The first step of PutA-catalyzing conversion of proline to glutamate requires FAD.

11. Figure 5, similar to the other EMSAs, it is not clear why PutA does not bind all of the probe DNA in these assays. Do the DNA probes in these experiments share a similar sequence to that of the promoter region for the *epsA* promoter?

Response: The probe concentration is high enough so PutA cannot bind all of the probe DNA in these EMSA experiments. The DNA probes in these experiments share a similar sequence to the binding sequence of PutA on the *epsA* promoter as shown in the following table (Fig. 5, Fig. S14).

Gene	The binding sequence
phcB	TTCCTTTTGCA
soll	AGGACATCCAG
trpEG	AGCGAAACGCA

12. Proline is reported to increase PutA-DNA binding whereas FAD and NAD⁺ are reported to decrease PutA binding to DNA. All of these factors are required for PutA enzymatic activity, what regulatory mechanism are the authors proposing? It is unclear. What about examining the ratio of NAD⁺/NADH, that would test whether PutA is a transcriptional redox sensor.

Response: It showed that the regulatory mechanism of PutA is a little complex. The PutA is not only involved in catalyzing proline but also works as a transcriptional regulator to control various gene expression (Fig. 4, Table S2). FAD and NAD⁺ promote the catalytic activity of PutA, while inhibit the regulatory activity of PutA. The structural analysis in the future should promote the complete elucidation of the regulatory mechanism of PutA, which needs further investigation.

13. What is the binding constant of PutA to the DNA? This is critical information and would help determine whether there is a priority of PutA binding to the different promoter regions of the target genes. Also, what change in binding affinity induced by proline or NAD⁺ would be sufficient to have meaningful regulatory effects?

Response: The previous structural analysis suggested that the hydrogen bonds between Lys9 and DNA promoter contribute to the transcriptional regulation of PutA in *E. coli* and thermodynamic data showed that the binding constant of PutA to the DNA is different in various buffer (Zhou et al., 2008, *J Mol Biol*). Our results showed that proline induces the regulatory activity of PutA while NAD⁺ and FAD inhibit it. It is difficult now to judge what is the priority of PutA binding, which needs further investigation.

14. Did the authors find any proline transporter genes in the global transcription analysis? It seems likely that PutA would also be involved in regulating proline transport.

Response: The previous study showed that PutP is involved in proline transportation in *E. coli* while there is no PutP or related proteins in *Ralstonia solanacearum*, suggesting that proline transporter in *R. solanacearum* is some different from that in *E. coli* (Jung et al, 2002, *FEBS Lett*).

Reviewers' comments:

Reviewer #1 (Remarks to the Author):

I believe that you have responded sincerely to the points I have raised and that the revised version provides stronger support for the conclusions of this paper. I am looking forward to reading this paper in *Communication Biology*.

Minor point

L258. I think the citation here should be 19, not 25.

Reviewer #2 (Remarks to the Author):

As far as my criticisms are concerned, the revision is ok for me.

Reviewer #3 (Remarks to the Author):

As stated in the previous review, the authors report interesting and important results showing PutA is involved in the pathogenesis of *R. solanacearum* in tomato plants. The main critique from the last review was a lack of a clear mechanism for how PutA/proline contribute to the observed effect on pathogenesis and whether PutA makes specific DNA binding interactions with the promoter regions. The responses provided by the authors do not sufficiently address concerns from the original review. The main issue for me still remains of whether the role of PutA in pathogenesis is due to its DNA binding activity. The evidence for PutA having specific binding interactions with the promoters is not convincing. This and other concerns are detailed below.

1. In response to point 6 from the original review, the authors added residue numbers to the PutA domain analysis in Figure 1a. Providing the residue numbers revealed a problem with the domain analysis shown in Fig. 1a. The DNA binding domain is incorrectly identified. The SMART database has not been updated properly. A conserved domain analysis using NCBI indicates a ribbon-helix-helix protein domain at the N-terminus, identifying PutA from *R. solanacearum* as part of the CopG family and PutA superfamily of proteins. The authors need to redo the domain analysis using another domain alignment tool.

2. The response to point 12 from the original review is not sufficient. The regulatory mechanism is still not adequately defined.

3. In regards to point 11, I disagree with the conclusion that "The probe concentration is high enough so PutA cannot bind all of the probe DNA in these EMSA experiments." In response to point 7, the authors state that the probe concentration is 3 ng/uL, which for 336 bp dsDNA, calculates to a concentration of ~14 nM. The concentrations of PutA used in the EMSA assays are 5-15 uM. I don't understand why a molar ratio of PutA/probe of ~350-1000 would not be sufficient for PutA to bind all of the probe DNA.

4. In response to point 11, the authors also state that there is sequence similarity of the PutA binding sites in the promoter regions. I can see a similar region between the *trpEG* (AGCGAAACGCA) and *epsA* (AGGGAAACGAA) promoters but not with *phcB* and *solI*. The similar binding regions in the promoters need to be clearly marked as they are not readily apparent.

5. Point 13, no DNA binding constant is provided for PutA. These experiments are needed to show that the affinity of PutA binding to the promoter region is consistent with it having a regulatory role.

6. The authors also point out that Lys9 is critical for DNA binding (response to point 13). This Lys is conserved in PutA from *R. solanacearum*. A suggested experiment to confirm that PutA is a global regulator would be to repeat the assays shown in Figure 1b-f, (and/or Fig 5f) with a PutA mutant (PutALys9Ala) that would be predicted to lack DNA binding activity. This would help distinguish between the DNA binding and metabolic roles of PutA. The inability of a PutALys9Ala

mutant to complement the putA mutant strain in these experiments, would indicate the DNA binding function of PutA is essential for pathogenesis.

7. Lines 453 and 468: Could the amount of protein-DNA complex be quantified from the 3 experiments and shown in the Figure?

8. Line 61- Supplement- Was the His-affinity tag removed? If so, this should be stated.

Reviewers' comments:

Reviewer #1 (Remarks to the Author):

I believe that you have responded sincerely to the points I have raised and that the revised version provides stronger support for the conclusions of this paper. I am looking forward to reading this paper in Communication Biology.

Response: Thank you very much for your valuable comments.

Minor point

L258. I think the citation here should be 19, not 25.

Response: We have revised it as suggested.

Reviewer #2 (Remarks to the Author):

As far as my criticisms are concerned, the revision is ok for me.

Response: Thank you very much for your valuable comments.

Reviewer #3 (Remarks to the Author):

As stated in the previous review, the authors report interesting and important results showing PutA is involved in the pathogenesis of *R. solanacearum* in tomato plants. The main critique from the last review was a lack of a clear mechanism for how PutA/proline contribute to the observed effect on pathogenesis and whether PutA makes specific DNA binding interactions with the promoter regions. The responses provided by the authors do not sufficiently address concerns from the original review. The main issue for me still remains of whether the role of PutA in pathogenesis is due to its DNA binding activity. The evidence for PutA having specific binding interactions with the promoters is not convincing. This and other concerns are detailed below.

1. In response to point 6 from the original review, the authors added residue numbers to the PutA domain analysis in Figure 1a. Providing the residue numbers revealed a problem with the domain analysis showed in Fig. 1a. The DNA binding domain is incorrectly identified. The SMART database has not been updated properly. A conserved domain analysis using NCBI indicates a ribbon-helix-helix protein domain at the N-terminus, identifying PutA from *R. solanacearum* as part of the CopG family and PutA superfamily of proteins. The authors need to redo the domain analysis using another domain alignment tool.

Response: Good suggestion, we have revised Fig.1a as suggested.

Domain structure analysis of PutA in *R. solanacearum* GMI1000 (bottom, HMMER, <https://www.ebi.ac.uk/Tools/hmmer/>), the red box represents the proline utilization A proline dehydrogenase N-terminal domain.

2. The response to point 12 from the original review is not sufficient. The regulatory mechanism is still not adequately defined.

Response: Yes, the regulatory mechanism of PutA is very interesting and complicated. Based on our findings, when the *R. solanacearum* GMI1000 colonized in the nutrition-limited environment in the vascular bundle of the host plant, the low concentration of NAD⁺ and FAD causes a reduced enzyme activity of PutA and results in a decrease of transformation of proline to glutamic acid, which then leads to the increase of proline concentration in the environment. The high concentration of proline enhances the regulatory activity of PutA to regulate the expression of pathogenic factors of *R. solanacearum* GMI1000. Of course, switch between PutA enzyme activity and regulatory activity is a complex process, which needs further comprehensive investigation.

3. In regards to point 11, I disagree with the conclusion that “The probe concentration is high enough so PutA cannot bind all of the probe DNA in these EMSA experiments.” In response to point 7, the authors state that the probe concentration is 3 ng/uL, which for 336 bp dsDNA, calculates to a concentration of ~14 nM. The concentrations of PutA used in the EMSA assays are 5-15 uM. I don’t understand why a molar ratio of PutA/probe of ~350-1000 would not be sufficient for PutA to bind all of the probe DNA.

Response: We first used the DNA probe at 3 ng/μg for the EMSA experiment. Then we cleaved the Tag from PutA as suggested in our revision of this manuscript and increased the DNA probe to 30 ng/μg. For 336 bp dsDNA, the concentration is calculated to be 135.2 nM ($C=10^6 * 30 / (660*336) = 135.2$ nM). The highest

concentration of PutA used in the EMSA assays is 15 μM ($C=4.5 \cdot 1000 / 147=30 \mu\text{M}$, and the final concentration of PutA is 15 μM), the molar ratio of PutA/probe is 111:1. As we know, different proteins bind to different probes in a different molar ratio, for example, the molar ratio of protein/probe is 48.7 μM :2nM, which is more than 20000:1, as shown in Fig 3 in the reference:

Bai, J. et al. The role of ArIRS in regulating oxacillin susceptibility in methicillin-resistant *Staphylococcus aureus* indicates it is a potential target for antimicrobial resistance breakers. *Emerg Microbes Infect* 8, 503-515 (2019).

4. In response to point 11, the authors also state that there is sequence similarity of the PutA binding sites in the promoter regions. I can see a similar region between the *trpEG* (AGCGAAACGCA) and *epsA* (AGGGAAACGAA) promoters but not with *phcB* and *solI*. The similar binding regions in the promoters need to be clearly marked as they are not readily apparent.

Response: The PutA binding site in the promoter region of *epsA* was identified to have the following characteristic sequence: AGGNAAANNA, *solI* possessed the similar sequence AGGACATCCAG, and the *phcB* (TTCCTTTTGCA) complemented sequence possessed the similar sequence AAGGAAAACGT.

5. Point 13, no DNA binding constant is provided for PutA. These experiments are needed to show that the affinity of PutA binding to the promoter region is consistent with it have a regulatory role.

Response: Good suggestion. We have added the experiments to analyze the affinity of PutA to the DNA promoter by microscale thermophoresis (MST) (Fig.2C and Fig. S16).

Fig.2

Fig. S16

6. The authors also point out that Lys9 is critical for DNA binding (response to point 13). This Lys is conserved in PutA from *R. solanacearum*. A suggested experiment to confirm that PutA is a global regulator would be to repeat the assays shown in Figure 1b-f, (and/or Fig 5f) with a PutA mutant (PutALys9Ala) that would be predicted to lack DNA binding activity. This would help distinguish between the DNA binding and metabolic roles of PutA. The inability of a PutALys9Ala mutant to complement the *putA* mutant strain in these experiments, would indicate the DNA binding function of PutA is essential for pathogenesis.

Response: Good suggestion. We have added these experiments and found that *in trans* expression of *putA*^{K9A} in the *putA* deletion mutant only partially restored these phenotypes, as shown in Fig.S3.

7. Lines 453 and 468: Could the amount of protein-DNA complex be quantified from the 3 experiments and shown in the Figure?

Response: It is not accurate to calculate the amount of protein-DNA by EMSA semi-quantitative method. We have measured the binding constants of protein-DNA by MST, which are shown in Fig.2C and Fig.S16.

8. Line 61- Supplement- Was the His-affinity tag removed? If so, this should be stated.

Response: Thank you, we have added more details in Supplement as suggested.

Reviewers' comments:

Reviewer #3 (Remarks to the Author):

The authors have provided additional experimental evidence in response to the previous review. The data with the putAK9A mutant show some partial role for PutA-DNA binding in the observed phenomenon. Additional data is also included in the revised manuscript to better quantitate the DNA binding activity of PutA.

Some other questions though still remain.

1. The authors do not provide any interpretation for the results with the putAK9A mutant. What does a partial restoration of the phenotypes mean? Do the authors know whether the K9A mutation results in loss of DNA binding activity?
2. The domain analysis shown in Figure 1b shows the DNA binding domain at residues 153-265. Why then would the K9A mutation be expected to disrupt DNA-binding activity of PutA? Did the authors try different structure analysis programs to confirm the correct assignment of the domains shown in Figure 1b?
3. The authors state that increased DNA probe was used for the binding experiments. I don't see this information about the probe concentrations mentioned anywhere in the manuscript or supplemental information. This should be provided.
4. The concentration of PutA that was held fixed for the MST experiments needs to also be provided.
5. A sequence alignment showing the similarity of the proposed PutA binding sites of the four promoters would be helpful.
6. Some other minor suggestions:
Line 108- "As a previous analysis ..." to "Previous analysis..."
Line 278 - change to "highly conserved"
Line 110- Change "To confirm the role" to "To test the role"

refer to Fig. 5B , Fig.6, Fig.7 in the reference mentioned above.

3. The authors state that increased DNA probe was used for the binding experiments. I don't see this information about the probe concentrations mentioned anywhere in the manuscript or supplemental information. This should be provided.

Response: Good suggestion, we have added more details in the Supplement Information as suggested.

4.The concentration of PutA that was held fixed for the MST experiments needs to also be provided.

Response: Good suggestion, the concentration of PutA is 50 nM, and we have added more details in the Supplement Information as suggested.

5. A sequence alignment showing the similarity of the proposed PutA binding sites of the four promoters would be helpful.

Response: Good suggestion. The PutA binding sites of the promoters were shown in the following picture.

6. Some other minor suggestions:

Line 108- "As a previous analysis ..." to "Previous analysis..."

Response: We have changed it as suggested.

Line 278 – change to "highly conserved"

Response: We have changed it as suggested.

Line 110- Change "To confirm the role" to "To test the role"

Response: We have changed it as suggested.